# An R2R3-MYB Transcription Factor OsMYBAS1 Promotes Seed Germination under Different Sowing Depths in Transgenic Rice

**DOI:** 10.3390/plants11010139

**Published:** 2022-01-05

**Authors:** Xiaomin Wang, Rong Wu, Tongshu Shen, Zhenan Li, Chengyong Li, Bangkui Wu, Hongye Jiang, Guangwu Zhao

**Affiliations:** 1The Key Laboratory for Quality Improvement of Agricultural Products of Zhejiang Province, College of Advanced Agricultural Sciences, Zhejiang Agriculture and Forestry University, Hangzhou 311300, China; wangxiaomin@zafu.edu.cn (X.W.); rongwu@stu.zafu.edu.cn (R.W.); shents233@stu.zafu.edu.cn (T.S.); Zhenanli@stu.zafu.edu.cn (Z.L.); wubangkui@stu.zafu.edu.cn (B.W.); redleafjiang@stu.zafu.edu.cn (H.J.); 2The Agro-Tech Extension Center of Quzhou, 139 Fushi Road, Quzhou 324000, China; Chengyongli@atecq.edu.cn

**Keywords:** antioxidant enzyme, deep-sowing, MYB transcription factor, OsMYBAS1, seed germination

## Abstract

MYB-type transcription factors play essential regulatory roles in seed germination and the response to seedling establishment stress. This study isolated a rice R2R3-MYB gene, *OsMYBAS1*, and functionally characterized its role in seed germination by generating transgenic rice plants with the overexpression and knockout of *OsMYBAS1*. Gene expression analysis suggested that *OsMYBAS1* was highly expressed in brown rice and root, respectively. Subcellular localization analysis determined that OsMYBAS1 was localized in the nucleus. No significant differences in seed germination rate were observed among wild-type (WT) and transgenic rice plants at the 0-cm sowing depth. However, when sown at a depth of 4 cm, higher germination rates, root lengths and seedling heights were obtained in *OsMYBAS1*-overexpressing plants than in WT. Furthermore, the opposite results were recorded between the *osmybas1* mutants and WT. Moreover, *OsMYBAS1*-overexpressing plants significantly enhanced superoxide dismutase (SOD) enzyme activity and suppressed the accumulation of malondialdehyde (MDA) content at the 4-cm sowing depth. These results indicate that the MYB transcription factor OsMYBAS1 may promote rice seed germination and subsequent seedling establishment under deep-sowing conditions. These findings can provide valuable insight into rice seed-quality breeding to facilitate the development of a dry, direct-seeding production system.

## 1. Introduction

Rice (*Oryza sativa* L.) is a staple food for more than half of the world’s population [1]. The demand for increased rice productivity is rising due to the increase in global food demand, climate change threats, water and land resources shortages, and the transfer of rural labor forces [2]. Compared with transplanting rice, which demands a large amount of water and energy input, the dry, direct seeding of rice is an efficient and labor-saving planting system, and it is more suitable for development by mechanization [3]. Previous studies have reported 15–20% higher grain yield and about 15% lower water use in dry, direct-seeded rice than transplanted-flooded rice [4,5]. However, some serious problems are often faced during dry, direct-seeded rice production. The surface sowing of seed is easily damaged by birds and mice and often results in non-uniform seedling establishment because of temperature variation [2,6]. To reduce the impact of these issues in the dry, direct-seeding system of rice, previous researchers have explored several corresponding agronomic measures to protect rice seed and establish uniform seedlings, including direct seeding at different sowing depths [2]. Unfortunately, deep sowing (more than 3 cm) adversely affects seed germination and seedling establishment, including increasing the time between seed germination and seedling emergence and growth in the mesocotyl length [7,8]. However, when encountering abiotic stress tolerance, such as in deep sowing, plants have evolved efficient mechanisms to sense and rapidly adapt to stressed conditions with many changes in physiological and molecular processes, respectively. The expression of stress-induced genes is primarily regulated by specific transcription factors [9]. Numerous studies have demonstrated that transgenic plants overexpressing genes encoding transcription factors can significantly enhance their tolerance to various abiotic stresses [10,11,12,13,14]. Therefore, it is essential to explore transcription factors regulating seed germination under different sowing depths, which can promote the sustainable development of a dry direct-seeding planting system.

It is well known that the MYB-type transcription factor family is present in all eukaryotes and MYB proteins play essential roles in multiple aspects of regulating responses to abiotic stress. The MYB gene was first identified as “Oncogene” *v-MYB* in avian myeloblastosis virus [15] and then a homologous gene, *Zea mays* C1, involved in the regulation of anthocyanin biosynthesis, was the first MYB gene to be characterized in plants [16]. Since, many MYB genes have been discovered in plants such as Arabidopsis and rice [9,12,17]. Based on the number and position of MYB DNA-binding domain repeats, the MYB protein family has been classified into four major groups, namely, 1R-MYB/MYB-related, R2R3-MYB, R1R2R3-MYB and 4R-MYB proteins [12,17]. R2R3-MYB transcription factors have a modular structure, with an N-terminal DNA-binding domain (the MYB domain) and an activation or repression domain, usually located at the C terminus. Moreover, R2R3-MYB proteins have been reported to be involved in the response of rice to abiotic stresses [10,14,18,19]. For instance, the overexpression of *OsMYB4* significantly confers tolerance to chilling and freezing stress in transgenic Arabidopsis [18,19]. *OsMYB2*-overexpressing rice lines exhibit enhanced tolerance to salt and cold, granted by a change in the expression levels of numerous genes involved in a diversity of functions in the stress response [10]. Moreover, *OsMYB6*-overexpressing rice increases tolerance to drought and salt stresses by increasing the activities of catalase (CAT) and SOD and suppressing the accumulation of MDA content [14]. In recent years, QTLs associated with deep-sowing tolerance have been detected in many chromosomes, and some studies have demonstrated that genes, including *PTOX1* and *OsTCP5*, are involved in deep-sowing tolerance by comparison of mutants and WT [8,20,21]. We previously found that *ZmMYB59*, an R2R3-MYB transcription factor, plays a negative regulatory role in seed germination under deep-sowing conditions [22]. However, knowledge is limited about the role of R2R3-MYB transcription factors of rice in regulating seed germination under different sowing depths. This study isolated an R2R3-MYB transcription factor, designated OsMYBAS1, in rice. The elucidation of *OsMYBAS1’s* function and regulation will provide the foundation for accelerating rice seed-quality breeding, which may facilitate the development of a dry, direct-seeding production system.

## 2. Results

### 2.1. Expression Profile of OsMYBAS1

The expression of *OsMYBAS1* in different rice tissues was monitored by qRT-PCR (Figure 1). The results showed that *OsMYBAS1* was expressed in glumes, panicle rachis branches, top second leaves, flag leaves, panicle rachises, sheaths, stems, brown rice and roots, respectively. Notably, the expression of *OsMYBAS1* in roots and brown rice was higher than in other tissues. Compared with the glume, the expression of *OsMYBAS1* in roots and brown rice was increased by 43.3-fold and 28.8-fold, respectively.

### 2.2. Subcellular Localization of OsMYBAS1

To examine the subcellular localization of *OsMYBAS1*, the recombinant constructs of the *OsMYBAS1-GFP* fusion gene and either of *GFP* or *mCherry* were introduced into rice protoplasts cells via PEG-Ca^2+^ mediated transformation. The results showed that the OsMYBAS1-GFP fusion protein and mCherry protein, alone, were explicitly localized in the nucleus, respectively, whereas GFP protein showed ubiquitous distribution in the whole cell (Figure 2).

### 2.3. OsMYBAS1 Regulates Seed Germination under Deep-Sowing Condition

To study the function of *OsMYBAS1*, an overexpressing construct and a CRISPR/Cas9 construct were transformed into the rice cultivar Nipponbare, and several transgenic lines were obtained (Figure 3A–C). Compared with WT, the expression of two *OsMYBAS1*-overexpressing lines, OE-1 and OE-2, were 17.8-fold and 8.1-fold higher, respectively. Moreover, two *osmybas1* mutants, *osmybas1-1* and *osmybas1-2*, were obtained by amplification of *Cas9* and *Hygromycin* and sequenced. As a result, one base, ‘G’, was added and five bases, ‘TAGCA’, were knocked out in *osmybas1-1,* while two bases, ‘AC’, were knocked out in *osmybas1-2*. 

To examine whether the phenotypes of transgenic lines differ from WT, homozygous T3 progeny of the transgenic lines and the WT were sown at depths of 0 and 4 cm, respectively (Figure 3D). Significant differences in phenotypes among WT, overexpressing plants and *osmybas1* mutants were observed under deep-sowing conditions (Table 1). When sown at a depth of 0 cm, no significant differences were observed in seed germination rate (95.3–97.3%) among WT, overexpressing plants and *osmybas1* mutants. Moreover, there were no distinct rules governing root length and seedling height changes, which ranged from 6.7 to 11.2 and 9.0 to 12.8 cm, respectively. However, the germination rates, root lengths and seedling heights of WT were significantly higher than *osmybas1* mutants and lower than overexpressing plants when sown at a depth of 4 cm. For instance, the germination rates of *osmybas1-1* and *osmybas1-2* were 32% and 35% lower than WT, while 24.3% and 31.3% higher than WT for OE-1 and OE-2, respectively.

### 2.4. Antioxidant Capacity of OsMYBAS1-Overexpressing Rice

To investigate whether *OsMYBAS1* expression influenced antioxidant capacity, the content of MDA and activities of CAT, POD and SOD were measured (Table 2). The MDA contents of overexpressing lines (OE-1 and OE-2) were significantly decreased, by 46.1–50.2% compared with WT lines. Moreover, the activities of SOD in the overexpression lines were enhanced considerably, by 11.4–17.9%, while no significant differences were observed in the activities of POD or CAT. These results imply that overexpression of *OsMYBAS1* confers a more efficient antioxidant system, counteracting oxidative stress under deep-sowing conditions.

## 3. Discussion 

The MYB proteins constitute one of the most prominent transcription factor families. In the rice genome, there are over 183 MYB-encoding genes, with diverse roles in abiotic stresses [23,24]. In this study, the expression of *OsMYBAS1* was found to be tissue-specific, showing the highest expression levels in roots, which is consistent with the results of Wang et al. [25]. However, *OsMYB2* was detected in different tissues and had the most significant expression in leaves, followed by roots and shoots [10]. Expression analysis of *OsMYB55* revealed that higher transcription was observed in the root tissues compared with the leaves, at the vegetative stages until tillering and the inflorescence stage. In contrast, its lowest expression level was recorded in all seeds’ development stages and maturation [26]. The highest expression of *OsMYB3R-2* was found in young stems, while the lowest was found in spikes [27]. Moreover, previous studies have determined that OsMYB2, OsMYB55 and OsMYB3R-2 play diverse functions in stress responses to salt, cold, higher temperature, freezing and drought. These results showed that the differential tissue expression patterns of *OsMYBAS1* in comparison with other MYB genes involved in abiotic stress tolerance indicate a possibility of a different role for it.

We analyzed its sequence of amino acids and found that the OsMYBAS1 protein contained two MYB DNA-binding domain repeats, which determines that OsMYBAS1 is a typical R2R3-MYB transcription factor. Previous studies have reported that numerous R2R3-MYB transcription factors are involved in the response and adaptation to abiotic stresses, such as seed germination and seedling establishment stresses [10,26]. Low germination rate and non-uniform seedling establishment have always been the primary factors restricting the high and stable yield of dry, direct-seeded rice. An appropriate increase in sowing depth could satisfy the requirements of soil temperature and moisture, which facilitate seed germination performance in a dry direct-seeding production system, but a sowing depth of more than 3 cm does not favor the seedling establishment and final yield [7]. Zhao et al. [8] found that the emergence rate of rice accessions was not significantly affected until the sowing depth reached 3 cm. Therefore, sowing depths deeper than 3 cm are efficient in evaluating the deep-sowing tolerance of rice. In this study, significant differences in germination rate, root length and seedling height were observed among WT, overexpressing plants and *osmybas1* mutants at a depth of 4 cm (Table 1). Moreover, *OsMYBAS1*-overexpressing plants enhanced seed germination compared with WT under deep-sowing conditions. These results indicate that OsMYBAS1 plays an essential role in mediating seed germination under deep-sowing conditions. This finding can provide valuable insight into rice seed-quality breeding to facilitate the development of a dry direct-seeding production system. 

Concentrations of reactive oxygen species (ROS) elevates and damages cellular structures, leading to the loss of germinating ability in abiotic stresses. Many studies have previously determined that MYB transcription factors enhance plants’ tolerance to various abiotic stresses by mitigating oxidative damage due to suppression of ROS production [10,12,14,28]. Yang et al. [10] found that *OsMYB2*-overexpressing plants increased their POD, SOD and CAT activities and had enhanced capacities for scavenging ROS under salt stress. Tang et al. [14] determined that the overexpression of *OsMYB6* in rice can increase tolerance to drought by increasing proline content, CAT and SOD activities, and decreased MDA content. In this study, overexpression of *OsMYBAS1* increased the activity of SOD and suppressed the accumulation of MDA content (Table 2), which can minimize oxidative damage. These results imply that the overexpression of *OsMYBAS1* confers a more efficient antioxidant system, counteracting oxidative stress under deep-sowing conditions.

The emergence of seedlings from the soil is associated with seed germination, shoot elongation, radicle elongation, hypocotyl elongation, seedling survival and soil conditions. Mu et al. [29] determined that the R2R3-MYB protein AtMYB59 was abundantly expressed in roots and regulated roots through binding to the downstream gene *CYCB1;1*. In this study, we found that the OsMYBAS1 protein was highly expressed in roots, and a higher root length was observed in *OsMYBAS1*-overexpressing lines. Moreover, we found that *AtMYB59* is highly homologous to *OsMYBAS1*. Thus, we speculate that the expression of OsMYBAS1 regulates root growth and enhances seed germination under deep-sowing conditions. Especially, the mesocotyl, coleoptile and first few internodes’ elongation are associated with seedling emergence when seeds are planted deeply [30,31,32]. Interestingly, we also found that increased hypocotyl elongation in the *OsMYBAS1*-overexpressing plants was observed, compared with WT plants under deep-sowing conditions. Yang et al. [2], Zhao et al. [8] and Lu et al. [33] found that increased mesocotyl length was induced by deep soil covering and was an essential characteristic in deep-sowing tolerance in the field. Previous studies have determined that some transcription factors, including ZmMYB59 and MYBH, regulate seed gemination under deep-sowing conditions by modulating hypocotyl elongation, which is involved in phytohormone signal pathways such as GA [22,34]. Moreover, we found that *ZmMYB59* is also highly homologous to *OsMYBAS1*. Therefore, we speculate that OsMYBAS1 enhanced seed germination associated with hypocotyl elongation via phytohormone signal pathway, and further studies need to confirm it. Notably, although *OsMYBAS1* is highly homologous to *ZmMYB59*, *OsMYBAS1* plays a positive role in seed germination under deep-sowing tolerance. It is well known that the MYB-type transcription factor family has many members, including *ZmMYB59* and *OsMYBAS1*, and MYB proteins play different roles in multiple aspects of regulating responses to abiotic stress. This phenomenon is ubiquitous and reported in previous studies [9,12]. Additionally, Smita et al. [13] observed ten guide-*OsMYBs* correlated with other *OsMYB* genes, forming a more complex feedback network, and found the presence of a feedback motif in the target OsMYBs. Comparing the top-down and guide-gene approaches, their results showed the conservation of one correlated pair of *OsMYB* (*OsMYBAS1* and *LOC_Os01g74410*). Notably, Zhao et al. [8] have reported that the candidate genes, *OsML1* and *OsML2,* for mesocotyl length were verified by an integrated analysis of GWAS, linkage mapping and allelic frequency differences. These studies provide valuable insight into elucidating the functions of OsMYBAS1. Hence, to thoroughly explain the underlying mechanisms of OsMYBAS1 regulating seed germination under deep-sowing conditions, further investigations should be conducted to demonstrate the potential target genes directly regulated by OsMYBAS1.

## 4. Material and Methods

### 4.1. Plant Material, Growth Conditions and Treatments 

Nipponbare (*Oryza sativa* L. ssp. *Japonica*) was used as WT for analysis of *OsMYBAS1* expression in different tissues, and was also exposed to different treatments. For the analysis of *OsMYBAS1* expression profiles in rice, the roots, leaves, stems, sheaths, panicle rachises, glumes and brown rice were sampled at harvest time and stored at −80 °C until further analysis. For sowing-depth treatment, seeds of rice cultivar Nipponbare, including WT, *OsMYBAS1*-overexpressing plants and *osmybas1* mutants, were sterilized in 75% ethanol for 3 min and in 20% NaClO for 30 min, and washed thoroughly with sterile water. The sterilized seeds were sown at a depth of 0 and 4 cm, respectively, with sterilized sand (diameter less than 0.8 mm and humidity between 60–70%), and then incubated in a growth chamber at 25 °C with 16-h light/8-h dark for ten days. All of the experiments contained three biological replications.

### 4.2. Subcellular Localization

The whole coding sequence of *OsMYBAS1* was ligated with NheI and XbaI-digested dual-luciferase reporter expression vector (detailed information of the vector referred to Gu et al. [35]) to generate 2×35S-*OsMYBAS1*-*GFP*. The GFP- and mCherry coding regions wee been inserted into the same expression vector, respectively, as control plasmids for protoplast transformation. The construct was confirmed by sequencing and used for the transient transformation of rice protoplast via PEG-Ca^2+^-mediated transformation. GFP and mCherry fluorescence in transformed rice cells were observed under a confocal microscope (LSM510, ZEISS, Jena, Germany).

### 4.3. Vector Construction and Plant Transformation 

The full-length cDNA of *OsMYBAS1* were amplified from rice with the primers 5′-GGGGTACCATGGTGACAGTGAGAGAGG-3′ and 5′-GGACTAGT TCATTTTCCATAACCAGATTG-3′. The product was ligated into the pGEM-T Easy vector and sequenced. Then, the *OsMYBAS1* fragment digested from pGEM-T Easy-*OsMYBAS1* was cloned into the KpnI-SpeI sites of a pEXT06 vector to obtain the pEXT06-*OsMYBAS1* construct. *OsMYBAS1* was driven by the cauliflower mosaic virus 35S (CaMV 35S) promoter. The pEXT06-*OsMYBAS1* construct was electroporated into *Agrobacterium tumefaciens* EHA105 and then introduced into the calli of the rice cultivar Nipponbare by *A. tumefaciens* EHA105-meditated methods. *OsMYBAS1* transgenic rice plants were selected in 1/2 MS medium containing 75 mg/L hygromycin (Roche, Mannheim, Germany). 

To generate the *osmybas1* mutants, the CRISPR/Cas9 editing system was utilized. In this system, *Cas9* was directed by the maize ubiquitin promoter, and the *sgRNA* expression cassette, which was directed by the OsU6 promoter, was arranged in tandem. For target recognition, 20-nt guide oligo-nucleotides (UP: 5′-TGTGTGGGTGGTCTAGGATAGCACGG-3′; LOW: 5′-AAACCCGTGCTATCCTAGACCACCCA-3′) were synthesized with appropriate adaptors for seamless ligation with the four promoters. The tandem *sgRNA* expression cassettes were first constructed in the PUC19 intermediate vector, and then sub-cloned into the PCAMBIA1300 backbone with the *Cas9* expression cassette to obtain the pCAMBIA1300-*OsMYBAS1*-*sgRNA*-*Cas9* construct. The construct was transformed into the calli of rice cultivar Nipponbare through Agrobacterium-mediated transformation.

### 4.4. qRT-PCR

Total RNA was isolated individually from the roots, leaves, stems, sheaths, panicle rachises, glumes and brown rice using TRIzol reagent. Reverse transcription was performed using a PrimeScript RT Enzyme Mix I based on the manufacturer’s instructions. qRT-PCR was performed in an optical 96-well plate with a CFX ConnectTM Real-Time system (BIO-RAD, Singapore). Each reaction contained 10 µL of 2 × Taq^Pro^ SYBR QPCR mix, 1 µL forward primer, 1 µL reverse primer, 5 µL cDNA and 3 µL ddH_2_O. The thermal cycle used was 95 °C for 30 s and 39 cycles of 95 °C for 5 s and 60 °C for 30 s, and 95 °C for 15 s. The primers of *OsMYBAS1* were 5′-GTGAACTACCTCCACCCTG-3′ (forward primer) and 5′-GCCTCCGTGCTATCCTA-3′ (reverse primer). Relative expression levels were calculated by the 2^−ΔΔCT^ method [36]. The amplification of *Actin* was used as an internal control to normalize data.

### 4.5. Germination Rate, Root Length and Seedling Height Measurements 

For the sowing-depth treatment, seeds of rice cultivar Nipponbare, including WT, *OsMYBAS1*-overexpressing plants and *osmybas1* mutants, were incubated in a growth chamber at 25 °C with a 16-h light/8-h dark. After 14 days, the germination rate, root length and seedling height were measured from each cultivar. Root length and seedling height were measured by the scale. The germination rate and was calculated according to the following formula:(1)$$\mathrm{germination}\;\mathrm{rate}\;(\%)=\frac{\mathrm{number}\;\mathrm{of}\;\mathrm{germinated}\;\mathrm{seeds}\;\mathrm{at}\;14\;\mathrm{days}}{\;\mathrm{total}\;\mathrm{number}\;\mathrm{of}\;\mathrm{seeds}}\;\times \;100$$

### 4.6. Determination of Peroxidase, Superoxide Dismutase and Catalase Activity and Malondialdehyde Content

The activities of peroxidase (POD; EC 1.11.1.7), SOD (EC 1.15.1.1) and CAT (EC 1.11.1.6), and MDA content were measured using methods described by Fu et al. [37].

### 4.7. Statistics 

Data were analyzed using the analysis of variance (ANOVA) procedure in SPSS 24.0 (IBM, Chicago, IL, USA), and multiple comparisons were explored using the Duncan test at a 0.05 probability level. Before data analysis, the percentage value was determined by arcsine transformation.

## 5. Conclusions

In this study, we identified a transcription factor, OsMYBAS1, that functions as a positive regulator to rice seed germination under deep-sowing conditions. Significant differences in germination rate, root length and seedling height were observed among WT, overexpressing plants and *osmybas1* mutants under deep-sowing conditions. Moreover, *OsMYBAS1*-overexpression rice increased the activity of SOD and suppressed the accumulation of MDA at the depth of 4 cm. These results enhance our understanding of the role of rice MYB transcription factors in the regulation of abiotic stress response and provide valuable insight into rice seed quality-breeding to facilitate the development of a dry direct-seeding production system.

## Figures and Tables

**Figure 1 plants-11-00139-f001:**
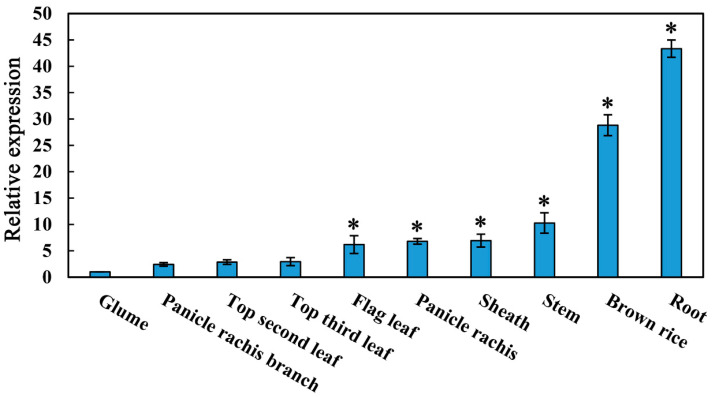
qRT-PCR analysis of the expression of *OsMYBAS1* in different rice tissues. Data are mean ± SE of three biological replications. Asterisks indicate statistically significant differences (*p* < 0.05, Duncan test) from the control (glume). *Actin* was used as an internal control.

**Figure 2 plants-11-00139-f002:**
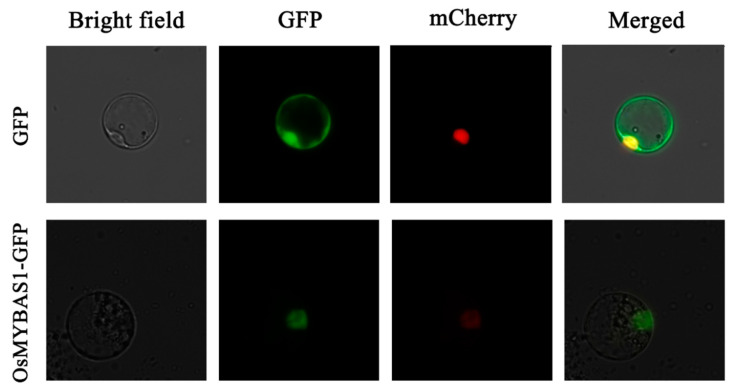
Subcellular localization analysis of *OsMYBAS1*. Confocal images of rice protoplasts cells under the GFP channel showing the constitutive localization of GFP and the nuclear localization of OsMYBAS1-GFP. Confocal images of rice protoplasts cells under the mCherry channel showing the constitutive localization of mCherry. The merged images of GFP and OsMYBAS1-GFP are presented, respectively.

**Figure 3 plants-11-00139-f003:**
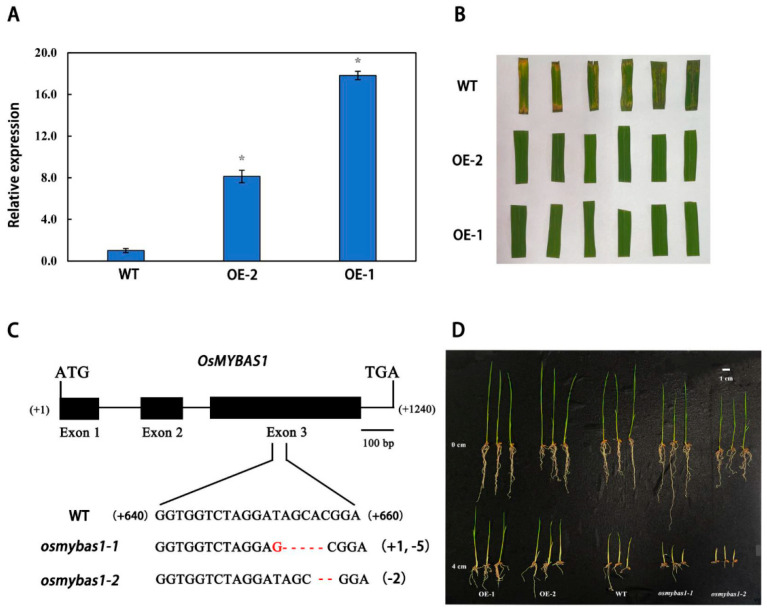
Identification and phenotype of *OsMYBAS1*-overexpressing plants and *osmybas1* mutants. (**A**) Relative expression of wild-type (WT) and two *OsMYBAS1*-overexpressing rice lines (OE-1 and OE-2); (**B**) leaf phenotype of WT and *osmybas1* mutants (*osmybas1-1* and *osmybas1-2*) soaked in Hygromycin solution; (**C**) two *osmybas1* mutants (*osmybas1-1* and *osmybas1-2*) were obtained by sequencing and the knockout sites are presented; (**D**) phenotype of WT, *OsMYBAS1*-overexpressing plants and *osmybas1* mutants at the sowing depths of 0 and 4 cm, respectively. Data are mean ± SE of three biological replications. Asterisks indicate statistically significant differences (*p* < 0.05, Duncan test) from the control (glume). *Actin* was used as an internal control.

**Table 1 plants-11-00139-t001:** Germination rates, root lengths and seedling heights of wild-type, *OsMYBAS1*-overexpressing plants and *osmybas1* mutants under different sowing depths.

Plant Material	0 cm Sowing-Depth			4 cm Sowing-Depth		
	Germination Rate (%)	Root Length (cm)	Seedling Height (cm)	Germination Rate (%)	Root Length (cm)	Seedling Height (cm)
WT	96.7 ± 3.3 a	7.9 ± 0.5 ab	11.7 ± 0.3 a	55.0 ± 3.8 b	2.9 ± 0.6 abc	4.8 ± 0.1 c
*osmybas1-1*	97.7 ± 1.5 a	11.2 ± 1.4 a	9.4 ± 0.8 b	23.0 ± 3.2 c	2.1 ± 0.6 bc	3.0 ± 0.3 d
*osmybas1-2*	96.7 ± 1.8 a	7.5 ± 1.1 b	9.0 ± 0.2 b	20.0 ± 2.5 c	1.5 ± 0.4 c	2.0 ± 0.1 d
OE-1	97.3 ± 1.5 a	9.0 ± 0.3 ab	12.0 ± 0.4 a	79.3 ± 1.8 a	4.5 ± 0.5 ab	8.3 ± 0.9 a
OE-2	95.3 ± 2.6 a	6.7 ± 1.4 b	12.8 ± 0.5 a	86.3 ± 2.0 a	3.9 ± 0.8 a	6.8 ± 0.1 b

Data are mean ± SE of three replications. Lower-cased letters indicate statistically significant differences (*p* < 0.05, Duncan test) between wild-type (WT) and transgenic plants.

**Table 2 plants-11-00139-t002:** Contents of oxidants and antioxidant enzymes in wild-type and *OsMYBAS1*-overexpressing plants.

Plant	SOD (U/g)	POD (nmol FW/min)	CAT (nmol FW/min)	MDA (nmol/g)
WT	731.3 ± 3.8c	23.4 ± 0.9a	1.0 ± 0.1a	415.8 ± 0.1a
OE-1	814.6 ± 0.8b	26.1 ± 0.2a	1.1 ± 0.4a	224.2 ± 0.2b
OE-2	861.8 ± 0.1a	26.1 ± 0.4a	1.4 ± 0.1a	207.1 ± 0.7c

Data are mean ± SE of three replicates. Lower-cased letters indicate statistically significant differences (*p* < 0.05, Duncan test) between wild-type (WT) and *OsMYBAS1*-overexpressing lines (OE-1 and OE-2). SOD, POD, CAT and MDA indicate superoxide dismutase, peroxidase, catalase and malondialdehyde, respectively.

## Data Availability

The data presented in this study are available in the graphs and tables provided in the manuscript.

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
