# Peer review of "An R2R3-MYB Transcription Factor OsMYBAS1 Promotes Seed Germination under Different Sowing Depths in Transgenic Rice"

_plants, 2022, doi:10.3390/plants11010139_

Round 1

Reviewer 1 Report

Manuscript by Wang et al. on “An R2R3-MYB Transcription Factor OsMYBAS1 Promotes Seed Germination under Different Sowing Depths in Transgenic Rice” studied the role of OsMYBAS1 in seedling establishment and development grown on surface and 4 cm depth. This study used wild type, overexpressing and knock out lines developed using CRISPR-Cas9 system and measured the effects on phenotypic traits. While the results from this study can be useful in developing strategies in combating abiotic stress in rice, following comments might be helpful in increasing the manuscript clarity.

Major comments:

The tissue-specific expression of the gene in this study differs to that of previous studies. How could that difference be observed for a given gene?

How was it concluded that suppressed MDA content and decreasing oxidative stress is helpful in beneficial way for seedling/plant establishment?

Were the edited-plants used in this study developed for the first time in this study? If yes, provide sequencing data for the confirmation of the knockouts. If not, cite the publication reporting the development of the mutants. Have these sequence data deposited in NCBI?

Minor comments:

Line 13: Replace * with # for equal contribution and confirm if correct.

Line 36: Be specific: “natural resources”

Line 39: Many studies but only two references cited?

Line 43: How does direct seeding favor uniform establishment? How is uniform establishment a serious problem?

Line 61: What is “it” referring to and replace?

Line 69: Include reference for “Moreover, R2R3-MYB …”

Line 82: Add role of in “limited about the role of R2R3-MYB transcription”

Line 84: Italicize OsMYBAS1 wherever applicable

Figure 1. This figure can be presented in black and white color (color-blind friendly). This figure can be a part of supplementary data because there is no comparison of presence/absence of the OsMYBAS1 gene.

Line 99: Remove the period at the end of the sub-title

Figure 2: Caption is not clear. Label each picture in alphabetical order and explain clearly what each image illustrates. GFP on y-axis indicates different variable compared to GFP on x-axis.

Line 118: Explain “… and sequence”

Line 119: were knocked out OR five-base sequence “TAGCA” was knocked out

Figure 3D. It is recommended that measuring scale be included in the image

Line 162: transcription factor

Line 166: Were leaves not included in shoots?

Line 186: If > 3 cm depth of sowing is not favorable for seedling establishment, why was 4 cm selected in this study?

Line 264: Confirm the primer sequence “5’-GGACTAGT TCATTTTCCA-…”

Line 284: Quantitative Real-time RT-PCR. It should be abbreviated as qRT-PCR in the entire manuscript wherever applicable. Total RNA extraction and cDNA synthesis procedures are missing.

Line 291: Confirm the qRT-PCR conditions: “95℃ for 30 s and 39 cycles of 95℃ for 5 s and 60℃ for 30 s, and 95℃ for 15 s”.

Line 296: Italicize subheading

Line 301: Although the methods have been previously described, provide brief description of the measurement procedure.

Line 308: Explain “percentage value was determined by arcsine transformation”. Were the data transformed? If yes, why was it needed?

Author Response

Dear Editor,

Thank you very much for your decision and encouragement, and we have revised our manuscript carefully according to the reviewers’ comments. The following is the response to the comments with a detailed and clear point-by-point reply. In the revised manuscript, the main revised and added parts are marked out by bold and red type.

Reviewer 1:

The following comments come from reviewer 1: Manuscript by Wang et al. on “An R2R3-MYB Transcription Factor OsMYBAS1 Promotes Seed Germination under Different Sowing Depths in Transgenic Rice” studied the role of OsMYBAS1 in seedling establishment and development grown on surface and 4 cm depth. This study used wild type, overexpressing and knock out lines developed using CRISPR-Cas9 system and measured the effects on phenotypic traits. While the results from this study can be useful in developing strategies in combating abiotic stress in rice, following comments might be helpful in increasing the manuscript clarity. Here we thanks for the encouragement from reviewer 1.

Comment 1: The tissue-specific expression of the gene in this study differs from that of previous studies. How could that difference be observed for a given gene?

Response: Thank you for your comments. We find the tissue-specific expression of OsMYBAS1 in NCBI (https://www.ncbi.nlm.nih.gov/gene/4351184), which involves in flower buds, flowers, flag leaves and roots sampled before flowering and after flowering, milk grains and mature seeds (Wang H, Niu QW, Wu HW, Liu J, Ye J, Yu N, Chua NH. Analysis of non-coding transcriptome in rice and maize uncovers roles of conserved lncRNAs associated with agriculture traits. Plant J. 2015 Oct;84(2):404-16. doi: 10.1111/tpj.13018. PMID: 26387578.). Wang et al. reported that the expression of OsMYBAS1 in roots was higher than in other tissues, which was consistent with our results. Moreover, we added the reference and incorporated related discussion (Lines 165-167, Page 5).

Comment 2: How was it concluded that suppressed MDA content and decreasing oxidative stress is helpful in beneficial way for seedling/plant establishment?

Response: Thank you for your comments. The increasement of reactive oxygen species (ROS) and MDA contents will damage cellular structures, which will lead to the loss of germinating ability and influence the seedling establishment. Many studies have determined that rice facing abiotic stresses will increase the reactive oxygen species (ROS) and MDA contents. Moreover, many studies previously have determined that MYB transcription factors enhance plants' tolerance to various abiotic stresses by mitigating oxidative damage due to suppression of ROS production, which is discussed in ‘Discussion’ section (Lines 198-210, Page 6). In this study, overexpression of OsMYBAS1 increased the activity of SOD and suppressed the accumulation of MDA content, which can minimize oxidative damage. These results imply that overexpression of OsMYBAS1 confers a more efficient antioxidant system to counteract oxidative stress under deep-sowing condition.

Comment 3: Were the edited-plants used in this study developed for the first time in this study? If yes, provide sequencing data for the confirmation of the knockouts. If not, cite the publication reporting the development of the mutants. Have these sequence data deposited in NCBI?

Response: Thank you for your comments. The edited-plants used in this study were developed for the first time. Two osmybas1 mutants (osmybas1-1 and osmybas1-2) were obtained by sequence and knockout sites were presented in figure 3 (Line 122, Page 4). The whole sequence of OsMYBAS1 (LOC4351184) can be obtained in NCBI (https://www.ncbi.nlm.nih.gov/gene/4351184). Moreover, we uploaded the file of sequencing data of mutants, which was not deposited in NCBI.

Comment 4: Line 13: Replace * with # for equal contribution and confirm if correct.

Response: Thank you for your comments. We have replaced * with # for equal contribution (Line 13, Page 1), and we also confirmed that.

Comment 5: Line 36: Be specific: “natural resources”.

Response: Thank you for your comments. Agriculture production mainly depends on natural resources. In China, water per capita is only 2,007 m3 and accounted for 25% of the world average. Moreover, two-thirds of the country's cities are short of water and a quarter is severely short of it. The effective utilization coefficient of farmland irrigation water is only 0.50, which is significantly lower than the world level.  Furthermore, per capita land resources are significantly lower than the world level and the area of arable land is decreasing in China. Therefore, water and land resources are short for agriculture production in China. We have replaced ‘natural resources’ with ‘water and land resources’ (Line 36, Page 1).

Comment 6: Line 39: Many studies but only two references cited?

Response: Thank you for your comments. We are sorry that the world is not precise, and the world ‘many’ has been revised as ‘previous’ (Line 39, Page 1). We want to emphasize Liu et al. and Wang et al. found that 15-20% higher grain yield and about 15% lower water use in dry direct-seeded rice than transplanted-flooded rice.

Comment 7: Line 43: How does direct seeding favor uniform establishment? How is uniform establishment a serious problem?

Response: Thank you for your comments. This is our negligence, and the world ‘uniform’ has been revised as ‘non-uniform’ (Line 43, Page 1). Direct seeding is easily damaged by birds and mice and faced with high/low temperatures, which often results in the non-uniform seedling establishment.

Comment 8: Line 61: What is “it” referring to and replace?

Response: Thank you for your comments. The world ‘it’ has been revised as ‘the MYB gene’ (Line 61, Page 2).

Comment 9: Line 69: Include reference for “Moreover, R2R3-MYB …”.

Response: Thank you for your comments. We have supplemented the related references (Line 71, Page 2).

Comment 10: Line 82: Add role of in “limited about the role of R2R3-MYB transcription”.

Response: Thank you for your comments. We have added role of in “limited about the role of R2R3-MYB transcription” with the sentence ‘However, knowledge is limited about the role of R2R3-MYB transcription factors of rice regulating seed germination under different sowing depths’ (Line 83, Page 2).

Comment 11: Line 84: Italicize OsMYBAS1 wherever applicable?

Response: Thank you for your comments. We have italicized OsMYBAS1 (Line 85, Page 2), and we carefully verified the manuscript.

Comment 12: Figure 1. This figure can be presented in black and white color (color-blind friendly). This figure can be a part of supplementary data because there is no comparison of presence/absence of the OsMYBAS1 gene.

Response: Thank you for your comments. In figure 1, only one index ‘relative expression’ was presented, and there is no ambiguity for people with color-blind. We also want to add color to make the picture more beautiful. Generally, the expression of one gene in different tissues was mesasured with wild-type plants. Therefore, the expression of OsMYBAS1 in rice different tissues was measured in wild-type. Moreover, we only want to determine the tissue expression patterns of OsMYBAS1.

Comment 13: Line 99: Remove the period at the end of the sub-title.

Response: Thank you for your comments. We have removed the period at the end of the sub-title.

Comment 14: Figure 2: Caption is not clear. Label each picture in alphabetical order and explain clearly what each image illustrates. GFP on y-axis indicates different variable compared to GFP on x-axis.

Response: Thank you for your comments. The figure 2 was the subcellular localization analysis, which refers to the specific location of a protein or the expression product of a gene in the cell. The regulation of gene expression is reflected in the structure and function of the protein. Only when the protein localization is correct, the normal biological function can be displayed. The eight pictures were a whole, and label each picture may be not suitable. We have suppled image illustrates (Lines 108-112, Page 3). Confocal images of rice protoplasts cells under the GFP channel showing the constitutive lo-calization of GFP and nuclear localization of OsMYBAS1-GFP. Confocal images of rice proto-plasts cells under the mCherry channel showing the constitutive localization of mCherry. The merged images of GFP and OsMYBAS1-GFP were presented, respectively.

Comment 15: Line 118: Explain “… and sequence”

Response: Thank you for your comments. We juded whether the knockout of OsMYBAS1 was successful through amplification of Cas9 and Hygromycin genes. If Cas9 or/and Hygromycin genes were presented in rice lines, it means that the knockout of OsMYBAS1 was failed. If Cas9 and Hygromycin genes were not amplificated in rice lines, and then we will measured the OsMYBAS1 sequences of the lines and ultimately determine the osmybas1 mutants.

Comment 16: Line 119: were knocked out OR five-base sequence “TAGCA” was knocked out.

Response: Thank you for your comments. We have revised the sentence ‘One base ‘G’ was added and five bases ‘TAGCA’ were knocked out in osmybas1-1 while two bases ‘AC’ was knockout in osmybas1-2’ (Lines 119-121, Page 4).

Comment 17: Figure 3D. It is recommended that measuring scale be included in the image.

Response: Thank you for your comments. The measuring scale was added at the upper right corner in figure 3D (Line 122, Page 4).

Comment 18: Line 162: transcription factor.

Response: Thank you for your comments. The world ‘transcript’ has been revised as ‘transcription’ (Line 163, Page 5).

Comment 19: Line 166: Were leaves not included in shoots?

Response: Thank you for your comments. In the study of OsMYB2 funtion, the shoot did not include leaves (Yang, A.; Dai, X.; Zhang, W.H. A R2R3-type MYB gene, OsMYB2, is involved in salt, cold, and dehydration tolerance in rice. J. Exp. Bot. 2012, 63, 2541-2556).

Comment 20: Line 186: If > 3 cm depth of sowing is not favorable for seedling establishment, why was 4 cm selected in this study?

Response: Thank you for your comments. At present, it is controversial that 3 cm depth of sowing influences the seedling establishment. Some studies found that the emergence rate of rice accessions was not affected greatly until the sowing depth reached 3 cm. 4 cm selected in this study can effiecently determine that the effecet of OsMYBAS1 on rice seed germination and subsequent seedling establishment under deep-sowing condition.

Comment 21: Line 264: Confirm the primer sequence “5’-GGACTAGT TCATTTTCCA-…”

Response: Thank you for your comments. We have confirmed the primer sequence.

Comment 22: Line 284: Quantitative Real-time RT-PCR. It should be abbreviated as qRT-PCR in the entire manuscript wherever applicable. Total RNA extraction and cDNA synthesis procedures are missing.

Response: Thank you for your comments. We have abbreviated ‘Quantitative Real-time RT-PCR’ in the entire manuscript wherever applicable. Total RNA was isolated using TRIzol reagent. Reverse transcription was performed using the PrimeScript RT Enzyme Mix I. Therefore, total RNA extraction and cDNA synthesis procedures could be referenced with the manufacturer’s in-structions.

Comment 23: Line 291: Confirm the qRT-PCR conditions: “95℃ for 30 s and 39 cycles of 95℃ for 5 s and 60℃ for 30 s, and 95℃ for 15 s”.

Response: Thank you for your comments. We have confirmed qRT-PCR conditions.

Comment 24: Line 296: Italicize subheading.

Response: Thank you for your comments. We have italicized subheading (Line 304, Page 8).

Comment 25: Line 301: Although the methods have been previously described, provide brief description of the measurement procedure .

Response: Thank you for your comments. We have provided brief description of the measurement procedure of root length, seedling height and germination rate (Lines 308-311, Page 8).

Comment 26: Line 308: Explain “percentage value was determined by arcsine transformation”. Were the data transformed? If yes, why was it needed?

Response: Thank you for your comments. Theoretically, both P value < 0.3 or P value > 0.7 need to perform arcsine transformation to improve the normality of the distribution, so as to obtain a more consistent variance. Before the analysis of ANOVA, we transformed the data with arcsine transformation. However, the percentage value should be presented in Tables, which reflected seed vigor.

Reviewer 2:

The following comments come from reviewer 2:The authors have presented a very interesting study on the functioning of a R2R3-MYB transcription factor OsMYBAS1 as a regulator of germination under varying sowing depth of the seedling. Here we thanks for the encouragement from reviewer 2.

Comment 1: Line 42-43: I think it will be 'non-uniform seedling establishment'.

Response: Thank you for your comments. This is our negligence, and the world ‘uniform’ has been revised as ‘non-uniform’ (Line 43, Page 1). Direct seeding is easily damaged by birds and mice and faced with high/low temperatures, which often results in the non-uniform seedling establishment.

Comment 2: As mentioned in the manuscript; 'ZmMYB59, an R2R3-MYB transcription factor, plays a negative regulatory role in seed germination under deep-sowing condition', 'ZmMYB59 is also highly homologous to OsMYBAS1' whereas as it seems from the results in case of rice it is a positive regulator. How would the authors explain this phenomenon and please incorporate this into the manuscript?

Response: Thank you for your comments. It is ubiquitous that houologous genes have different roles for abiotic stress tolerance. Moreover, we have incorporated explain into the manuscript with the sentence ‘Notably, although OsMYBAS1 is highly homologous to ZmMYB59, OsMYBAS1 plays a positive role for seed germination under deep-sowing tolerance. It is well known that the MYB-type transcription factor family has many members including ZmMYB59 and OsMYBAS1, and MYB proteins play different roles in multiple aspects of regulating responses to abiotic stress. This phenomenon is ubiquitous and reported in previous studies [9,12]’ at Lines 231-236, Page 7.

Comment 3: After reading this paper the readers will be left with a question on how the overexpression and mutant lines of OsMYBAS1 respond to abiotic stresses like drought or salinity. Can the authors incorporate a seedling level physiological study on this aspect into the manuscript?

Response: Thank you for your comments. In this study, our results indicate that the MYB transcription factor OsMYBAS1 may promote rice seed germination and subsequent seedling establishment under deep-sowing condition. Moreover, we also measured physiological indexs, and OsMYBAS1-overexpressing plants significantly enhanced the activity of the superox-ide dismutase (SOD) enzyme and suppressed the accumulation of malondialdehyde (MDA) con-tent at the 4 cm sowing depth and we incorporated discussion of a seedling level physiological study in ‘Discussion’ section (Lines 202-210, Page 6). However, many studies need to be conducted to determine the regulatory role of OsMYBAS1 under drought or salinity.

We are looking forward to your positive decision.

Yours sincerely,

Guangwu Zhao,

College of Advanced Agricultural Sciences,

Zhejiang Agriculture and Forestry University,

Lin’an 311300, Zhejiang, China

Reviewer 2 Report

The authors have presented a very interesting study on the functioning of a R2R3-MYB transcription factor OsMYBAS1 as a regulator of germination under varying sowing depth of the seedling. Still I have few comments mentioned underneath.

Line 42-43: I think it will be 'non-uniform seedling establishment'

As mentioned in the manuscript; 'ZmMYB59, an R2R3-MYB transcription factor, plays a negative regulatory role in seed germination under deep-sowing condition', 'ZmMYB59 is also highly homologous to OsMYBAS1' whereas as it seems from the results in case of rice it is a positive regulator. How would the authors explain this phenomenon and please incorporate this into the manuscript?

After reading this paper the readers will be left with a question on how the overexpression and mutant lines of OsMYBAS1 respond to abiotic stresses like drought or salinity. Can the authors incorporate a seedling level physiological study on this aspect into the manuscript?

Author Response

Dear Editor,

Thank you very much for your decision and encouragement, and we have revised our manuscript carefully according to the reviewers’ comments. The following is the response to the comments with a detailed and clear point-by-point reply. In the revised manuscript, the main revised and added parts are marked out by bold and red type.

Reviewer 1:

The following comments come from reviewer 1: Manuscript by Wang et al. on “An R2R3-MYB Transcription Factor OsMYBAS1 Promotes Seed Germination under Different Sowing Depths in Transgenic Rice” studied the role of OsMYBAS1 in seedling establishment and development grown on surface and 4 cm depth. This study used wild type, overexpressing and knock out lines developed using CRISPR-Cas9 system and measured the effects on phenotypic traits. While the results from this study can be useful in developing strategies in combating abiotic stress in rice, following comments might be helpful in increasing the manuscript clarity. Here we thanks for the encouragement from reviewer 1.

Comment 1: The tissue-specific expression of the gene in this study differs from that of previous studies. How could that difference be observed for a given gene?

Response: Thank you for your comments. We find the tissue-specific expression of OsMYBAS1 in NCBI (https://www.ncbi.nlm.nih.gov/gene/4351184), which involves in flower buds, flowers, flag leaves and roots sampled before flowering and after flowering, milk grains and mature seeds (Wang H, Niu QW, Wu HW, Liu J, Ye J, Yu N, Chua NH. Analysis of non-coding transcriptome in rice and maize uncovers roles of conserved lncRNAs associated with agriculture traits. Plant J. 2015 Oct;84(2):404-16. doi: 10.1111/tpj.13018. PMID: 26387578.). Wang et al. reported that the expression of OsMYBAS1 in roots was higher than in other tissues, which was consistent with our results. Moreover, we added the reference and incorporated related discussion (Lines 165-167, Page 5).

Comment 2: How was it concluded that suppressed MDA content and decreasing oxidative stress is helpful in beneficial way for seedling/plant establishment?

Response: Thank you for your comments. The increasement of reactive oxygen species (ROS) and MDA contents will damage cellular structures, which will lead to the loss of germinating ability and influence the seedling establishment. Many studies have determined that rice facing abiotic stresses will increase the reactive oxygen species (ROS) and MDA contents. Moreover, many studies previously have determined that MYB transcription factors enhance plants' tolerance to various abiotic stresses by mitigating oxidative damage due to suppression of ROS production, which is discussed in ‘Discussion’ section (Lines 198-210, Page 6). In this study, overexpression of OsMYBAS1 increased the activity of SOD and suppressed the accumulation of MDA content, which can minimize oxidative damage. These results imply that overexpression of OsMYBAS1 confers a more efficient antioxidant system to counteract oxidative stress under deep-sowing condition.

Comment 3: Were the edited-plants used in this study developed for the first time in this study? If yes, provide sequencing data for the confirmation of the knockouts. If not, cite the publication reporting the development of the mutants. Have these sequence data deposited in NCBI?

Response: Thank you for your comments. The edited-plants used in this study were developed for the first time. Two osmybas1 mutants (osmybas1-1 and osmybas1-2) were obtained by sequence and knockout sites were presented in figure 3 (Line 122, Page 4). The whole sequence of OsMYBAS1 (LOC4351184) can be obtained in NCBI (https://www.ncbi.nlm.nih.gov/gene/4351184). Moreover,  we will upload the file of sequencing data of mutants, which was not deposited in NCBI.

Comment 4: Line 13: Replace * with # for equal contribution and confirm if correct.

Response: Thank you for your comments. We have replaced * with # for equal contribution (Line 13, Page 1), and we also confirmed that.

Comment 5: Line 36: Be specific: “natural resources”.

Response: Thank you for your comments. Agriculture production mainly depends on natural resources. In China, water per capita is only 2,007 m3 and accounted for 25% of the world average. Moreover, two-thirds of the country's cities are short of water and a quarter is severely short of it. The effective utilization coefficient of farmland irrigation water is only 0.50, which is significantly lower than the world level.  Furthermore, per capita land resources are significantly lower than the world level and the area of arable land is decreasing in China. Therefore, water and land resources are short for agriculture production in China. We have replaced ‘natural resources’ with ‘water and land resources’ (Line 36, Page 1).

Comment 6: Line 39: Many studies but only two references cited?

Response: Thank you for your comments. We are sorry that the world is not precise, and the world ‘many’ has been revised as ‘previous’ (Line 39, Page 1). We want to emphasize Liu et al. and Wang et al. found that 15-20% higher grain yield and about 15% lower water use in dry direct-seeded rice than transplanted-flooded rice.

Comment 7: Line 43: How does direct seeding favor uniform establishment? How is uniform establishment a serious problem?

Response: Thank you for your comments. This is our negligence, and the world ‘uniform’ has been revised as ‘non-uniform’ (Line 43, Page 1). Direct seeding is easily damaged by birds and mice and faced with high/low temperatures, which often results in the non-uniform seedling establishment.

Comment 8: Line 61: What is “it” referring to and replace?

Response: Thank you for your comments. The world ‘it’ has been revised as ‘the MYB gene’ (Line 61, Page 2).

Comment 9: Line 69: Include reference for “Moreover, R2R3-MYB …”.

Response: Thank you for your comments. We have supplemented the related references (Line 71, Page 2).

Comment 10: Line 82: Add role of in “limited about the role of R2R3-MYB transcription”.

Response: Thank you for your comments. We have added role of in “limited about the role of R2R3-MYB transcription” with the sentence ‘However, knowledge is limited about the role of R2R3-MYB transcription factors of rice regulating seed germination under different sowing depths’ (Line 83, Page 2).

Comment 11: Line 84: Italicize OsMYBAS1 wherever applicable?

Response: Thank you for your comments. We have italicized OsMYBAS1 (Line 85, Page 2), and we carefully verified the manuscript.

Comment 12: Figure 1. This figure can be presented in black and white color (color-blind friendly). This figure can be a part of supplementary data because there is no comparison of presence/absence of the OsMYBAS1 gene.

Response: Thank you for your comments. In figure 1, only one index ‘relative expression’ was presented, and there is no ambiguity for people with color-blind. We also want to add color to make the picture more beautiful. Generally, the expression of one gene in different tissues was mesasured with wild-type plants. Therefore, the expression of OsMYBAS1 in rice different tissues was measured in wild-type. Moreover, we only want to determine the tissue expression patterns of OsMYBAS1.

Comment 13: Line 99: Remove the period at the end of the sub-title.

Response: Thank you for your comments. We have removed the period at the end of the sub-title.

Comment 14: Figure 2: Caption is not clear. Label each picture in alphabetical order and explain clearly what each image illustrates. GFP on y-axis indicates different variable compared to GFP on x-axis.

Response: Thank you for your comments. The figure 2 was the subcellular localization analysis, which refers to the specific location of a protein or the expression product of a gene in the cell. The regulation of gene expression is reflected in the structure and function of the protein. Only when the protein localization is correct, the normal biological function can be displayed. The eight pictures were a whole, and label each picture may be not suitable. We have suppled image illustrates (Lines 108-112, Page 3). Confocal images of rice protoplasts cells under the GFP channel showing the constitutive lo-calization of GFP and nuclear localization of OsMYBAS1-GFP. Confocal images of rice proto-plasts cells under the mCherry channel showing the constitutive localization of mCherry. The merged images of GFP and OsMYBAS1-GFP were presented, respectively.

Comment 15: Line 118: Explain “… and sequence”

Response: Thank you for your comments. We juded whether the knockout of OsMYBAS1 was successful through amplification of Cas9 and Hygromycin genes. If Cas9 or/and Hygromycin genes were presented in rice lines, it means that the knockout of OsMYBAS1 was failed. If Cas9 and Hygromycin genes were not amplificated in rice lines, and then we will measured the OsMYBAS1 sequences of the lines and ultimately determine the osmybas1 mutants.

Comment 16: Line 119: were knocked out OR five-base sequence “TAGCA” was knocked out.

Response: Thank you for your comments. We have revised the sentence ‘One base ‘G’ was added and five bases ‘TAGCA’ were knocked out in osmybas1-1 while two bases ‘AC’ was knockout in osmybas1-2’ (Lines 119-121, Page 4).

Comment 17: Figure 3D. It is recommended that measuring scale be included in the image.

Response: Thank you for your comments. The measuring scale was added at the upper right corner in figure 3D (Line 122, Page 4).

Comment 18: Line 162: transcription factor.

Response: Thank you for your comments. The world ‘transcript’ has been revised as ‘transcription’ (Line 163, Page 5).

Comment 19: Line 166: Were leaves not included in shoots?

Response: Thank you for your comments. In the study of OsMYB2 funtion, the shoot did not include leaves (Yang, A.; Dai, X.; Zhang, W.H. A R2R3-type MYB gene, OsMYB2, is involved in salt, cold, and dehydration tolerance in rice. J. Exp. Bot. 2012, 63, 2541-2556).

Comment 20: Line 186: If > 3 cm depth of sowing is not favorable for seedling establishment, why was 4 cm selected in this study?

Response: Thank you for your comments. At present, it is controversial that 3 cm depth of sowing influences the seedling establishment. Some studies found that the emergence rate of rice accessions was not affected greatly until the sowing depth reached 3 cm. 4 cm selected in this study can effiecently determine that the effecet of OsMYBAS1 on rice seed germination and subsequent seedling establishment under deep-sowing condition.

Comment 21: Line 264: Confirm the primer sequence “5’-GGACTAGT TCATTTTCCA-…”

Response: Thank you for your comments. We have confirmed the primer sequence.

Comment 22: Line 284: Quantitative Real-time RT-PCR. It should be abbreviated as qRT-PCR in the entire manuscript wherever applicable. Total RNA extraction and cDNA synthesis procedures are missing.

Response: Thank you for your comments. We have abbreviated ‘Quantitative Real-time RT-PCR’ in the entire manuscript wherever applicable. Total RNA was isolated using TRIzol reagent. Reverse transcription was performed using the PrimeScript RT Enzyme Mix I. Therefore, total RNA extraction and cDNA synthesis procedures could be referenced with the manufacturer’s in-structions.

Comment 23: Line 291: Confirm the qRT-PCR conditions: “95℃ for 30 s and 39 cycles of 95℃ for 5 s and 60℃ for 30 s, and 95℃ for 15 s”.

Response: Thank you for your comments. We have confirmed qRT-PCR conditions.

Comment 24: Line 296: Italicize subheading.

Response: Thank you for your comments. We have italicized subheading (Line 304, Page 8).

Comment 25: Line 301: Although the methods have been previously described, provide brief description of the measurement procedure .

Response: Thank you for your comments. We have provided brief description of the measurement procedure of root length, seedling height and germination rate (Lines 308-311, Page 8).

Comment 26: Line 308: Explain “percentage value was determined by arcsine transformation”. Were the data transformed? If yes, why was it needed?

Response: Thank you for your comments. Theoretically, both P value < 0.3 or P value > 0.7 need to perform arcsine transformation to improve the normality of the distribution, so as to obtain a more consistent variance. Before the analysis of ANOVA, we transformed the data with arcsine transformation. However, the percentage value should be presented in Tables, which reflected seed vigor.

Reviewer 2:

The following comments come from reviewer 2:The authors have presented a very interesting study on the functioning of a R2R3-MYB transcription factor OsMYBAS1 as a regulator of germination under varying sowing depth of the seedling. Here we thanks for the encouragement from reviewer 2.

Comment 1: Line 42-43: I think it will be 'non-uniform seedling establishment'.

Response: Thank you for your comments. This is our negligence, and the world ‘uniform’ has been revised as ‘non-uniform’ (Line 43, Page 1). Direct seeding is easily damaged by birds and mice and faced with high/low temperatures, which often results in the non-uniform seedling establishment.

Comment 2: As mentioned in the manuscript; 'ZmMYB59, an R2R3-MYB transcription factor, plays a negative regulatory role in seed germination under deep-sowing condition', 'ZmMYB59 is also highly homologous to OsMYBAS1' whereas as it seems from the results in case of rice it is a positive regulator. How would the authors explain this phenomenon and please incorporate this into the manuscript?

Response: Thank you for your comments. It is ubiquitous that houologous genes have different roles for abiotic stress tolerance. Moreover, we have incorporated explain into the manuscript with the sentence ‘Notably, although OsMYBAS1 is highly homologous to ZmMYB59, OsMYBAS1 plays a positive role for seed germination under deep-sowing tolerance. It is well known that the MYB-type transcription factor family has many members including ZmMYB59 and OsMYBAS1, and MYB proteins play different roles in multiple aspects of regulating responses to abiotic stress. This phenomenon is ubiquitous and reported in previous studies [9,12]’ at Lines 231-236, Page 7.

Comment 3: After reading this paper the readers will be left with a question on how the overexpression and mutant lines of OsMYBAS1 respond to abiotic stresses like drought or salinity. Can the authors incorporate a seedling level physiological study on this aspect into the manuscript?

Response: Thank you for your comments. In this study, our results indicate that the MYB transcription factor OsMYBAS1 may promote rice seed germination and subsequent seedling establishment under deep-sowing condition. Moreover, we also measured physiological indexs, and OsMYBAS1-overexpressing plants significantly enhanced the activity of the superox-ide dismutase (SOD) enzyme and suppressed the accumulation of malondialdehyde (MDA) con-tent at the 4 cm sowing depth and we incorporated discussion of a seedling level physiological study in ‘Discussion’ section (Lines 202-210, Page 6). However, many studies need to be conducted to determine the regulatory role of OsMYBAS1 under drought or salinity.

We are looking forward to your positive decision.

Yours sincerely,

Guangwu Zhao,

College of Advanced Agricultural Sciences,

Zhejiang Agriculture and Forestry University,

Lin’an 311300, Zhejiang, China

Round 2

Reviewer 1 Report

I would like to thank Wang et al. for revising the manuscripts and answering the reviewers' queries. I do not have any further comments. However, I would request authors to correct the grammatical errors that are present in the manuscript.

Author Response

Dear Editor,

Thank you very much for your decision and encouragement, and we have revised our manuscript carefully according to the reviewer’s comments. The following is the response to the comments with a detailed and clear point-by-point reply. In the revised manuscript, the main revised and added parts are marked out by bold and red type.

Reviewer:

Comment: I would like to thank Wang et al. for revising the manuscripts and answering the reviewers' queries. I do not have any further comments. However, I would request authors to correct the grammatical errors that are present in the manuscript.

Response: Thank you for your comments. We carefully checked the English language and corrected grammatical errors. The main modifications are as follows:

(1) the sentence ‘at the depth of 4 cm’ has been revised as ‘at a depth of’ (Line 21; Line 133; Line 136; Line 141; Line 194; Line 259, respectively).

(2) the sentence ‘OsMYBAS1-overexpressing plants significantly enhanced the activity of the superoxide dismutase (SOD) enzyme’ has been revised as ‘OsMYBAS1-overexpressing plants significantly enhanced the superoxide dismutase (SOD) enzyme activity (Lines 23-25).

(3) the word ‘condition’ has been revised as ‘conditions’ (Line 27; Line 83; Line 135; Line 156; Line 195; Line 196; Line 211; Line 220; Line 224; Line 228; Line 246; Line 324; Line 326, respectively).

(4) the sentence ‘productivity of rice’ was revised as ‘rice productivity’ (Line 35).

(5) we removed the sentence ‘as well as’ (Line 36).

(6) the sentence ‘the labor force’ was revised as ‘the transfer of rural labor force’ (Lines 36-37).

(7) the sentence ‘an increase in’ was revised as ‘increasing’ (Line 49).

(8) the sentence ‘an increase’ was revised as ‘a growth’ (Line 49).

(9) the sentence ‘is largely regulated’ was revised as ‘is primarily regulated’ (Line 53).

(10) we added the sentence ‘as a result’ (Line 120).

(11) the sentence ‘Moreover, there were no distinct rules in the changes of root length and seedling height’ was revised as ‘Moreover, there were no distinct rules in root length and seedling height changes’ (Lines 137-138).

(12) the sentence ‘significantly enhanced’ was revised as ‘enhanced considerably’ (Line 153).

(13) the sentence ‘Affected greatly’ was revised as ‘significantly affected’ (Line 190).

(14) the word ‘found’ was revised as ‘finding’ (Line 198).

(15) the sentence ‘Comparing the results from both top-down and guide-gene approaches showed’ was revised as ‘Comparing the top-down and guide-gene approaches, results showed’ (Lines 241-242).

We are looking forward to your positive decision.

Yours sincerely,

Guangwu Zhao,

College of Advanced Agricultural Sciences,

Zhejiang Agriculture and Forestry University,

Lin’an 311300, Zhejiang, China
